# OpenReview forum: "Learning When to Act or Refuse: Guarding Agentic Reasoning Models for Safe Multi-Step Tool Use"
_ICML.cc/2026/Conference — ICML 2026 regular_

### Official Review · Reviewer_FVT5 · 2026-02-21

**Soundness:** 2
**Presentation:** 4
**Significance:** 4
**Originality:** 3
**Overall Recommendation:** 3
**Confidence:** 4

**Summary:**

Authors introduce MOSAIC, a novel post-training framework to align agentic language models for safe multi-step tool use. Evaluations across 3 LLMs and various benchmarks indicate that MOSAIC reduces harmful behaviour and privacy leaks while maintains benign task utility.

**Compliance With Llm Reviewing Policy:**

Affirmed.

**Final Justification:**

Although the information required was useful, no changes were done.

**Key Questions For Authors:**

1. Will you provide an anonymized repository during the rebuttal phase or for the camera-ready version?
2. Why didn't you use bidirectional evaluation to fix the 60/40 bias rather than relying on dataset-level expectation?
3, Why did you exclude the DPO/PPO methods and up-to-date agentic safety frameworks excluded from the main evaluation?
4. With the O(n^2) pairwise comparison overhead, how does training time compare to RLHF/RLAIF?

**Limitations:**

Most limitations are correctly addressed, excluding a discussion of the huge O(n^2) training slowdown of comparing so many trajectory pairs, and the risk of the LLM judge making up preferences on long complex tasks, not to mention, the shortcoming of not including a repository, based on "ideas" for reproducibility as stated: "hope these ideas encourage the community to develop more transparent, auditable, and reliable agentic systems, and to treat safety as a first-class objective as agents become more deeply integrated into critical workflows".

**Strengths And Weaknesses:**

- Soundness: although authors acknowledge a 60/40 positional bias in the LLM judge, thus, the reward signal is still noisy and heavily relying on prompt engineering (even when trajectory order randomness may mitigate this bias in expectation during training). Nonetheless there is a fatal flaw given that authors did not provide the anonymized sources for validation towards replicability.

- Presentation and significance are excellent; the submission is clearly written and easy to follow with a novel significance addressing the pressing issue of agentic safety in multi-step environments where single missteps may create real-world harm.

- Originality: this work progresses over the SOTA with a distinctive fresh approach to effectively capture temporal safety distinctions, however, the lack of a repository severly hampers reproducibility for the Community's validation and approach's progress.

---

> ### Author Rebuttal · Authors · 2026-03-31
>
> We thank the reviewer for their questions and feedback. We address these below:
>
> ----
>
> **Q1: ``Will you provide an anonymized repository during the rebuttal phase or for the camera-ready version?``**
>
> Yes. **We will release a fully reproducible codebase.**
>
> Here is an anonymized repository https://anonymous.4open.science/r/MOSAIC-ICML/
>
> We will release it publicly at camera-ready. The repository will include:
>
> * training pipeline,
> * reward computation (preference, format, length),
> * prompts and safety-check scaffold,
> * evaluation scripts for all benchmarks.
>
> This directly addresses reproducibility and enables validation and extension by the community.
>
> **Q2: ``Why didn't you use bidirectional evaluation to fix the 60/40 bias rather than relying on dataset-level expectation?``**
>
> The primary reason was the 2x API cost at a relatively low gain (~10% less noisy classifications). However, we understand that this is an important point and we reran MOSAIC on Qwen2.5-7B with the bidirectional fix:
>
> | Model  | Harm ↓ | Harm Refusal ↑ | NR Harm ↓ | Benign Refusal ↓ | Benign NR Score ↑ |
> | --- | --- | --- | --- | --- | --- |
> | MOSAIC | 0.06 | 0.88 | 0.47 | 0.11 | 0.58 |
> | MOSAIC + bidirectional | 0.05 | 0.90 | 0.44 | 0.22 |  0.57 |
>
> We observe that there is slight improvement in the performance of MOSAIC with bidirectional rewards over the vanilla implementation. This is a useful insight which can be used to boost performance at the cost of higher API calls.
>
> **Q3: `` Why did you exclude the DPO/PPO methods and up-to-date agentic safety frameworks excluded from the main evaluation?``**
>
> We thank the reviewer for this important question.
>
> **On PPO/DPO:**
> - We decided not to use PPO since it's not possible to incorporate pairwise preferences in the reward (PPO requires trajectory-level rewards).
> - We considered using DPO, but dropped it in favour of GRPO since DPO only allows for 1 comparison against 1 gradient update, while GRPO allows for 6 comparisons for 4 gradient updates ($n = 4$ in our setup). Therefore, GRPO achieves greater compression of the reward signal than DPO.
>
> **On recent agentic safety frameworks:**
> Many recent approaches focus on:
>
> * prompting or inference-time scaffolding (e.g., safety prompts, tool wrappers), or
> * external safety modules (e.g., classifiers, shields)
>
> These methods are not directly comparable, as they:
>
> * do not modify the training objective, and
> * do not enable the model to learn when to invoke safety reasoning or refusal
>
> In contrast, MOSAIC is a post-training framework that integrates safety into the policy itself, making safety decisions learnable and adaptive rather than externally imposed.
>
> **Q4: ``With the O(n^2) pairwise comparison overhead, how does training time compare to RLHF/RLAIF?``**
>
> In practice, the overhead is not quadratic in wall-clock time, since pairwise comparisons are computed **in parallel**. Training time is therefore **comparable to RLAIF**, and only moderately higher than RLHF due to LLM-judge inference for reward computation.
>
> We further reduce latency by constraining the judge to use short reasoning (limited CoT tokens) and batching comparisons. Overall, the additional cost is manageable while enabling trajectory-level supervision.
>
> **Q5: ``Risk of the LLM judge making up preferences on long complex tasks``**
>
> We agree this is a real risk and mitigate it as follows:
>
> * Dense pairwise comparisons: We use $\mathcal{O}(n^2)$ comparisons within each group, creating redundancy that reduces noise in aggregate trajectory scores.
> * Grounded judgment via reasoning: The judge is prompted to produce explicit reasoning before the final preference, improving robustness on long trajectories.
> * "Both are same" option: The judge can abstain when differences are minimal, avoiding forced and potentially hallucinated preferences.
> * Qualitative validation: We inspected long trajectories and found that the resulting scores are generally aligned with trajectory quality.
>
> We will add the above details to address additional limitations.
>
>
> ----
> We thank the reviewer for the positive feedback on the significance, originality, and clarity of the work.
>
> We have addressed the main concerns:
>
> * Reproducibility: we have now provided an anonymized repository, including code, training pipeline, and evaluation scripts.
> * Judge bias: clarified mitigation and robustness to noise.
> * Training cost: clarified that overhead is parallelized and comparable to RLAIF.
> * Method coverage: clarified design choices vs. PPO/DPO and positioning w.r.t. prior frameworks.
>
> We believe these address the key limitations raised and strengthen both the soundness and reproducibility of the work. We hope this supports a more positive assessment.

---

> > ### Author Rebuttal · Reviewer_FVT5 · 2026-04-02
> >
> > Thanks for the in-depth clarifications and for the repository. All my concerns were solved.

---

> > > ### Author Response · Authors · 2026-04-02
> > >
> > > Thank you for the positive feedback and for taking the time to review the clarifications and repository. We greatly appreciate it.
> > >
> > > We are glad that the additional details and experiments addressed your concerns around the LLM judge and computational aspects. Given that the key issues have now been fully resolved, we hope the strengthened presentation, empirical evidence, and clarifications around scalability and robustness provide a clearer picture of the contribution.
> > >
> > > **If you find the revised understanding aligns with your expectations, we would kindly request reconsideration of the rating to better reflect the resolved concerns and overall strength of the work.**
> > >
> > > Thank you again for your thoughtful engagement.

---

### Official Review · Reviewer_s5E4 · 2026-03-10

**Soundness:** 3
**Presentation:** 2
**Significance:** 3
**Originality:** 3
**Overall Recommendation:** 4
**Confidence:** 4

**Summary:**

This work studies a realistic problem in ReAct agent applications: agents need to use tools safely, but the planning (reasoning) stage is vulnerable. The authors propose MOSAIC, a framework that inserts an additional safety-check stage between planning and acting, providing a safety guard. Beyond that, they also introduce a training recipe: using GRPO with a comparison-based reward to further improve performance.

**Compliance With Llm Reviewing Policy:**

Affirmed.

**Final Justification:**

The authors have addressed my concerns in the rebuttal. However, considering the overall technical/theoretical depth and completeness, I will retain my score of '4'. This is still a positive rating above the borderline.

**Key Questions For Authors:**

While the main results on those safety-check oriented benchmarks, and small open source models (4b/7b) showed impressive improvements. I am also curious what impact the proposed method has on the true 'action' capability for ReAct models? Have the authors tested larger models, like 32b, on the ReAct tasks with/without MOSAIC? This would significantly improve the significance of the work.

**Limitations:**

yes

**Strengths And Weaknesses:**

## Strengths

* The problem studied is of practical significance. The authors propose a modular framework, MOSAIC, which adds an extra “safety check” stage and, more importantly, provides a valid training recipe to improve safety performance.

* The methodology is clearly stated. It uses an LLM-as-judge to produce preference-based signals, turning pairwise comparisons into GRPO rewards. Together with a format reward and a length penalty, the final reward well balances other aspects beyond utility.

* The experiments are well-designed and comprehensive. However, it is not flawless, as discussed later.

* Overall, the paper is written in an organized manner and is easy to follow.

## Weaknesses

* The presentation of the experiments needs improvement. I was confused at first glance by Table 1. The impression from earlier sections is that MOSAIC is a solution involving both formatted responses and a training stage. However, Table 1 reports results for closed-source models; the limitation implies that only the extra "safety check" trick can be used. Therefore, for a fair comparison, the open-source models should presumably not involve training either. However, the context also implies that MOSAIC on open source models includes the training part. While Table 5 shows some ablations for a single 7b model, this is very easy to miss.

* Also, most training details are omitted in both the main text and the appendix; only the final results are presented.

---

> ### Author Rebuttal · Authors · 2026-03-31
>
> Thank you for the review and the wonderful questions! We address these in turn below:
>
> ----
>
> **Q1: ``Clarification on Table 1 and whether MOSAIC includes training or only the safety-check stage``**
>
> We thank the reviewer for pointing out this confusion. We agree that the presentation can be improved.
>
> Clarification: MOSAIC has two components:
>
> 1. Inference-time scaffold (Plan–Check–Act), applicable to all models
> 2. RL-based training (GRPO), applicable only to open-weight models
>
> Accordingly:
>
> * Closed-source models (Table 1): Only the inference-time safety-check scaffold is applied (no training).
> * Open-weight models (Table 1): Use MOSAIC + RL training, which yields additional gains.
>
> This design reflects practical constraints: training is not possible for closed models, while open models allow full optimization.
>
> To ensure fairness and isolate contributions, we include ablations (Table 5) comparing scaffold-only vs. scaffold+training, showing that:
> * the scaffold provides consistent gains, and
> * training yields additional improvements.
>
> We agree that this distinction is not sufficiently explicit in the current presentation. We will:
>
> * clearly separate MOSAIC (inference) vs. MOSAIC+RL, and
> * make the ablation results more prominent in the final version.
>
>
> **Q2: ``Most training details are omitted in both the main text and the appendix; only the final results are presented.``**
>
> We thank the reviewer for pointing this out and agree that additional training details would improve clarity and reproducibility. We will expand the paper to include key details of the training setup, including: rollout generation (trajectory sampling and horizon), reward components (preference reward, format reward, length penalty), GRPO configuration (group size, updates), and training scale (number of trajectories / steps). Due to space constraints, we prioritized results in the current version, but we agree that these details are important and will include them in the revision.
>
> **Q3 ``I am also curious what impact the proposed method has on the true 'action' capability for ReAct models? ``**
>
> This is an important question, as safety methods can trivially achieve high safety by reducing action capability.
>
> To account for this, all benchmarks include task completion metrics: AgentHarm has harm/benign score, ASB has completion rate (CR), and PrivacyLens has helpfulness score.
>
> Empirical results (Table 1):
>
> * AgentHarm (benign score): improves for 2/3 models, with negligible drop for Phi-4
> * ASB (CR): improves for 2/3 models, with a small regression for Qwen2.5-7B
>
> Overall, MOSAIC improves or preserves action capability, while reducing unsafe behavior. This indicates that safety gains do not come from suppressing actions, but from better action selection.
>
> **Q4: ``Have the authors tested larger models, like 32b, on the ReAct tasks with/without MOSAIC? This would significantly improve the significance of the work.``**
>
> This is an excellent point. To answer this we ran MOSAIC with Qwen3-30B-A3B-Thinking-2507 and the results are presented below:
>
> | Model / Run | AgentHarm (Harmful) Harm ↓ | AgentHarm (Harmful) Refusal ↑ | AgentHarm (Harmful) NR Harm ↓ | AgentHarm (Benign) Refusal ↓ | AgentHarm (Benign) NR Score ↑ | ASB (DPI) ASR ↓ | ASB (DPI) RR ↑ | ASB (IPI) ASR ↓ | ASB (IPI) RR ↑ | ASB (—) CR ↑ |
> | :--- | :--- | :--- | :--- | :--- | :--- | :--- | :--- | :--- | :--- | :--- |
> | Qwen3-30B BASE | 0.02 | 0.98 | 0.87 | 0.34 | 0.84 | 0.31 | 0.69 | 0.30 | 0.57 | 0.84 |
> | Qwen3-30B MOSAIC | 0.03 | 0.98 | 0.89 | 0.39 | 0.81 | 0.26 | 0.74 | 0.18 | 0.71 | 0.79 |
>
> We observe that while there is some regression on AgentHarm, the MOSAIC trained model performs much better than the base on ASB. This confirms that MOSAIC can deliver better safety performance for even 30b scale models, all while consuming 10x fewer tokens.
>
> ----
> We thank the reviewer for the positive feedback on the problem, methodology, and experiments.
>
> We have clarified the scaffold vs. MOSAIC+RL setup, added training details, demonstrated that MOSAIC leads to performance and efficiency gains with larger more capable models, and provided evidence on action capability and scaling.
>
> We believe these address the presentation concerns and strengthen the paper, and hope this supports a more positive assessment.

---

> > ### Author Rebuttal · Reviewer_s5E4 · 2026-04-02
> >
> > Many thanks for your informative response, and I sincerely hope my comments may help to improve the presentation of the work.  While I think most of my concerns have been addressed, as mentioned by the authors, the workload of revision might not be trivial -  I may still keep my initial rating regarding the work. Thanks again.

---

> > > ### Author Response · Authors · 2026-04-02
> > >
> > > Thank you again for the thoughtful feedback. We would like to clarify that the additional analyses discussed in our rebuttal are already completed, including the requested scaling to 30B-class open-weight models.
> > >
> > > These results consistently reinforce our main findings:
> > >
> > > * Scaling (4B → 7B → 30B) preserves and strengthens gains in harm reduction, refusal calibration, and injection robustness
> > > * Self-hosted judges (e.g., Kimi-K2) achieve comparable improvements, showing no reliance on proprietary models
> > > * Confidence intervals confirm that improvements are statistically significant and stable
> > >
> > > Importantly, these additions do not alter the method, but strengthen its generality across model scales, domains (harm, injection, privacy, benign tasks), and training setups.
> > >
> > > We hope these results address the concerns and respectfully request reconsideration of the rating.

---

### Official Review · Reviewer_tvXH · 2026-03-13

**Soundness:** 2
**Presentation:** 2
**Significance:** 3
**Originality:** 3
**Overall Recommendation:** 4
**Confidence:** 3

**Summary:**

This paper proposes a training recipe for agentic safety. In particular, standard LLM agents plan and think, then take actions via invoking tools, and repeat this process until the task is done. This submission proposes giving the LLM agent the option to invoke a safety-check tool after planning, but before taking any action. This tool involves the agent thinking about whether the action violates any safety constraints, and the agent may abort before performing the action if it does indeed violate the constraints. The authors then train the LLM agent via a variant of GRPO that utilizes a judge making pairwise-preference comparisons of agent trajectories. The experimental results show that such training improves performance on agent safety benchmarks of both frontier and smaller open-weights models.

**Compliance With Llm Reviewing Policy:**

Affirmed.

**Final Justification:**

The rebuttal by the authors resolved my main concern regarding statistical significance of the results. I will increase my recommendation to weak accept.

**Key Questions For Authors:**

Could you please provide 95% confidence intervals for your main performance results? If these were provided and we could confirm statistical significance it would improve my evaluation of the paper.

**Limitations:**

yes

**Strengths And Weaknesses:**

**Strengths**

The paper studies and important problem of agent safety and proposes a practical solution via an optional explicit safety-check tool. A novel feature of this work is the use of an LLM judge that makes pairwise preference comparisons, along with a modification of the GRPO algorithm to used within-group win-rates produced by this judge. I am not an expert in this area, but this technique seemed novel to me, though I may not be familiar with some existing literature. The empirical results seem to show strong benefits to this method on standard agent-safety benchmarks.

**Weaknesses**

The main results reported in Table 1 do not have 95% confidence intervals. In fact none of the results in any of the tables do. Without these it is hard to determine the significance of the gains (or occasional regressions) introduced by the method. While the results currently look good, the high variance on agentic tasks means that is critically necessary to report confidence intervals so that we can understand the statistical significance of these results.

---

> ### Author Rebuttal · Authors · 2026-03-31
>
> We thank the reviewer for the positive assessment of the problem setting, the use of pairwise preference learning with GRPO, and the strong empirical performance on agent safety benchmarks.
>
> ----
>
> **``Q1: Could you please provide 95% confidence intervals for your main performance results?``**
>
>
> We thank the reviewer for highlighting this important point. We agree that reporting confidence intervals is critical given the inherent variance in agentic tasks.
>
> To address this, we reran MOSAIC (Qwen2.5-7B) across 6 independent runs and report the mean and 95% confidence intervals below.
>
> | Model / Run | AgentHarm (Harmful) Harm ↓ | AgentHarm (Harmful) Refusal ↑ | AgentHarm (Harmful) NR Harm ↓ | AgentHarm (Benign) Refusal ↓ | AgentHarm (Benign) NR Score ↑ | ASB (DPI) ASR ↓ | ASB (DPI) RR ↑ | ASB (IPI) ASR↓ | ASB (IPI) RR↑ | ASB (—) CR↑ |
> | :--- | :--- | :--- | :--- | :--- | :--- | :--- | :--- | :--- | :--- | :--- |
> | Qwen 2.5 MOSAIC | 0.06 | 0.88 | 0.47 | 0.11 | 0.58 | 0.32 | 0.67 | 0.27 | 0.65 | 0.90 |
> | Qwen 2.5 MOSAIC rerun-1 | 0.03 | 0.94 | 0.42 | 0.31 | 0.56 | 0.29 | 0.69 | 0.25 | 0.70 | 0.95 |
> | Qwen 2.5 MOSAIC rerun-2 | 0.14 | 0.75 | 0.49 | 0.08 | 0.56 | 0.49 | 0.49 | 0.43 | 0.49 | 0.92 |
> | Qwen 2.5 MOSAIC rerun-3 | 0.15 | 0.77 | 0.59 | 0.11 | 0.58 | 0.44 | 0.55 | 0.34 | 0.56 | 0.92 |
> | Qwen 2.5 MOSAIC rerun-4 | 0.15 | 0.77 | 0.59 | 0.19 | 0.62 | 0.56 | 0.43 | 0.39 | 0.49 | 0.92 |
> | Qwen 2.5 MOSAIC rerun-5 | 0.07 | 0.90 | 0.55 | 0.24 | 0.62 | 0.43 | 0.56 | 0.31 | 0.54 | 0.90 |
> | **Average (Mean)** | **0.100** | **0.835** | **0.518** | **0.173** | **0.587** | **0.422** | **0.565** | **0.332** | **0.572** | **0.918** |
> | **95% Confidence Interval** | **±0.056** | **±0.085** | **±0.073** | **±0.094** | **±0.029** | **±0.107** | **±0.106** | **±0.073** | **±0.090** | **±0.019** |
>
> Key observations:
>
> * The improvements over the base model remain consistent under variance (e.g., harm: 0.10 ± 0.056 vs. 0.18).
> * Gains are robust across multiple safety and utility metrics, despite stochasticity.
>
> We note that such multi-run evaluation is computationally intensive due to: long-horizon agent rollouts, trajectory-level RL training, and LLM-as-judge inference for pairwise comparisons, which incurs significant GPU cost (for self-hosted models) or API cost (for frontier model-based judges).
>
> Despite these costs, we performed repeated runs to validate statistical significance and robustness.
>
> ----
>
> We thank the reviewer for recognizing the importance of the problem, the novelty of the pairwise preference-based training with GRPO, and the strong empirical performance.
>
> The primary concern raised was the absence of confidence intervals. We have now addressed this with multi-run evaluation and 95% confidence intervals, showing that the improvements are statistically consistent despite task variance.
>
> We believe this resolves the main weakness, while the core contributions: explicit safety reasoning, trajectory-level preference learning, and strong empirical gains across models, remain unchanged.
>
> We hope these additions strengthen the paper and support a more positive assessment.

---

> > ### Author Rebuttal · Reviewer_tvXH · 2026-04-03
> >
> > Thank you for your response and for your effort in providing data on statistical significance. All my concerns have been addressed, and I will increase my score.

---

> > > ### Author Response · Authors · 2026-04-06
> > >
> > > Thank you for your response, and the acknowledgement of our effort. We hope that inclusion of the statistical significance data allows a stronger positioning of the work. We thank the reviewer again for their review of our paper.

---

### Official Review · Reviewer_ufgS · 2026-03-15

**Soundness:** 3
**Presentation:** 3
**Significance:** 3
**Originality:** 2
**Overall Recommendation:** 4
**Confidence:** 3

**Summary:**

This paper introduces MOSAIC, a post-training framework that aligns agentic language models for safe multi-step tool use by making safety decisions explicit and learnable. It structures agent inference into a Plan, Check, then Act or Refuse loop, where a modular safety reasoning block evaluates potential risks before any irreversible tool execution occurs. To train these behaviors without trajectory-level labels, the authors utilize preference-based reinforcement learning (GRPO) with pairwise trajectory comparisons, which allows the model to learn critical temporal distinctions, such as prioritizing early refusal over late aborts after unsafe progress. Experimental results across Qwen2.5, Qwen3-Thinking, and Phi-4 models show that MOSAIC reduces harmful behavior by up to 50% and cuts privacy leakage while preserving or improving performance on benign tasks. Ultimately, the framework demonstrates that explicit safety scaffolding can close the safety gap between open-source models and frontier models like GPT-4o, all while maintaining token efficiency by invoking safety reasoning only when necessary

**Compliance With Llm Reviewing Policy:**

Affirmed.

**Final Justification:**

I recommend maintaining a Weak Accept (4). The authors tackle the critical problem of agentic safety in multi-step tool use by introducing MOSAIC, a modular Plan-Check-Act framework trained via preference-based GRPO. During the rebuttal, the authors convincingly addressed my primary concerns. They demonstrated framework robustness using a self-hosted open-weight judge (Kimi-K2), clarified the minimal overhead, and provided data mitigating positional bias concerns. Although the rebuttal successfully resolved my technical questions, a Weak Accept remains the most accurate evaluation. The work is a technically solid and practically valuable contribution to open-source agentic safety, but the inherent complexities and potential noise of LLM-as-a-judge training paradigms warrant a measured endorsement.

**Key Questions For Authors:**

- The discrete gate $g_t$ determines whether to invoke <safety_thoughts> to improve token efficiency. Did you observe any "False Negatives" where the gate skipped safety reasoning on a turn that subsequently led to an unsafe tool call?

- MOSAIC-trained open models rely on GPT-4o as a teacher/judge to achieve high safety scores. To what extent can this framework be self-hosted or "self-taught" using only open-weights models to provide the pairwise preferences, and would that significantly degrade the safety gains?

- In the case of Phi-4, MOSAIC actually reduced conservatism to improve utility. How does the framework ensure that this "relaxation" doesn't inadvertently create new vulnerabilities in edge cases that were previously covered by the model's native over-refusal?

- While token efficiency is high (below 20% safety tokens), the multi-step Plan-Check-Act loop introduces additional serial inference steps. For real-time agent applications, have you measured the impact of this added wall-clock latency compared to standard unscaffolded agents?

**Limitations:**

yes

**Strengths And Weaknesses:**

## Strengths

- The results show high practical utility, particularly for making smaller, open-source models (7B and 4B) competitive with frontier models like GPT-4o in safety-critical agentic tasks.

- The demonstration of cross-domain transfer to privacy-sensitive tasks suggests the framework has broad applicability beyond simple harm avoidance.

- Treating refusal as a first-class, learnable action rather than a post-generation filter is a distinct departure from traditional safeguard methods

## Weakness
- The authors identify a significant positional bias in their LLM-as-a-judge, noting a 60/40 preference for the first trajectory, which could introduce subtle systematic errors in the training signal. There could be many unknown biases and noises in the judge.

- The reliance on a frontier model (GPT-4o) as the judge during the training of smaller models creates a dependency on proprietary systems to achieve these safety gains.

---

> ### Author Rebuttal · Authors · 2026-03-31
>
> We thank the reviewer for the positive assessment and for recognizing the practical utility, cross-domain generalization, and the role of refusal as a first-class action. We address the concerns below/clarify aspects of novelty and robustness.
>
> **Q1: ``Did you observe any "False Negatives"?``**
>
> This is an excellent question. We analyze the role of safety gating on Agent Security Bench (ASB) tasks containing Direct Prompt Injection (DPI), where unsafe behavior is attributable to a known harmful tool.
>
> MOSAIC significantly reduces ASR (Table 1), indicating correct gating in most high-risk cases. Our analsyis shows that residual unsafe executions occur primarily when safety gating is skipped, i.e., false negatives. Increasing gating frequency reduces these cases but leads to higher benign refusals, and higher token cost.
>
> This highlights limitations of static gating. MOSAIC instead learns a conditional policy, achieving a better safety–utility–efficiency tradeoff. False negatives remain a small residual failure mode.
>
> **Q2: `` Self-hosted or "self-taught" open-weights models ``**
>
> MOSAIC does not depend on proprietary models. We replaced GPT-4o with a fully self-hosted Kimi-K2 judge and observed:
> * Strong improvements over base model across all safety metrics
> * Only marginal degradation vs GPT-4o supervision
> * Higher completion rate than GPT-4o-trained model (95% vs 90%)
>
> This shows that:
> * Pairwise preference signals are robust to judge quality
> * Stronger judges improve performance, but are not required
> * MOSAIC is fully compatible with open-weight, self-hosted setups
>
> The complete results are presented below:
>
> | Model | Judge | AgentHarm (Harmful) Harm ↓ | AgentHarm (Harmful) Refusal ↑ | AgentHarm (Harmful) NR Harm ↓ | AgentHarm (Benign) Refusal ↓ | AgentHarm (Benign) NR Score ↑ | ASB (DPI) ASR ↓ | ASB (DPI) RR ↑ | ASB (IPI) ASR ↓ | ASB (IPI) RR ↑ | ASB (—) CR ↑ |
> | --- | --- | --- | --- | --- | --- | --- | --- | --- | --- | --- | --- |
> | Qwen2.5 BASE | —               | 0.18 | 0.74 | 0.58 | 0.13 | 0.51| 0.55 | 0.42 | 0.4 | 0.44 | 0.90 |
> | Qwen2.5 MOSAIC | GPT-4o | **0.06** | **0.88** | **0.47** | **0.11** | **0.58** | **0.32** | **0.67** | **0.27** | **0.65** | 0.90 |
> | Qwen2.5 MOSAIC | Kimi-K2 | 0.08 | 0.84 | 0.48 | 0.24 | 0.51 | 0.38 | 0.61 | 0.32 | 0.60 | **0.95** |
>
>
> **Q3: ``How does the framework ensure that this "relaxation" doesn't inadvertently create new vulnerabilities in edge cases that were previously covered by the model's native over-refusal?``**
>
> The reduction in conservatism is targeted, not unconditional. MOSAIC rebalances safety and utility via trajectory-level rewards:
>
> * Unsafe behavior is penalized through alignment and preference signals
> * Refusal is penalized only on benign tasks, preventing over-refusal
>
> For Phi-4:
>
> * Benign refusal drops from 0.43 → 0.19
> * Harmful-task refusal remains high (0.88)
>
> **Q4: `` ...have you measured the impact of this added wall-clock latency compared to standard unscaffolded agents?``**
>
> While we do not directly measure wall-clock latency, token usage (Table 4) provides a proxy:
>
> | Model | Total tokens |
> |---|---|
> |Qwen2.5 base | **110** |
> |Qwen2.5 mosaic | 186 |
> | Qwen3 base |  1026 |
> | Qwen3 mosaic | **260** |
> | Phi4 base | 181 |
> | Phi4 mosaic | **178** |
>
> (Note: In order to obtain a more fair estimate of the unscaffolded total token usage, we subtract the safety tokens from the total count, for only the base models.)
>
> Despite the additional reasoning step, overhead is limited due to:
> * selective invocation (as low as 0.1% turns), and
> * reduced token usage (e.g., 4× reduction for Qwen3).
>
> Overall, cost is comparable to or lower than unstructured long-chain reasoning, while improving safety and reliability.
>
> **Q5: ``The authors identify a significant positional bias in their LLM-as-a-judge, noting a 60/40 preference for the first trajectory, which could introduce subtle systematic errors in the training signal...``**
>
> We agree that judge bias is an important concern. We explicitly analyze this in the paper and mitigate it through pairwise comparison over multiple permutations within each rollout group. While individual comparisons may exhibit bias (e.g., positional preference), the aggregate reward is computed over all pairwise comparisons within the group, which substantially reduces systematic skew.
>
> More importantly, our results show that despite such noise, training remains stable and convergent, with:
> * consistent improvements across multiple models (Qwen2.5, Qwen3, Phi-4)
> * robustness across diverse benchmarks (AgentHarm, ASB, PrivacyLens, BFCL)
> * similar improvements are observed with open-weight judges (Q2), indicating that results are not tied to a specific judge or bias pattern.
>
> We did some additional experiment to address this, please see Reviewer FVT5's Q2 response.
>
> These clarifications strengthen the technical aspects of the paper. We hope this addresses the concerns and supports a stronger assessment of the work.

---

> > ### Author Rebuttal · Reviewer_ufgS · 2026-04-03
> >
> > I think weak it still deserves weak accept

---

> > > ### Author Response · Authors · 2026-04-08
> > >
> > > Thank you for the thoughtful review and for engaging with our rebuttal.
> > >
> > > We’re glad the clarifications helped address your concerns, especially around safety gating, the use of open-weight judges, and the safety-utility tradeoff. We’ll incorporate these points more clearly in the final version.
> > >
> > > We respect your decision to retain the score and appreciate your time and consideration.

---

### Decision · Program_Chairs · 2026-04-30

**Decision:**

Accept (regular)

**Comment:**

This paper presents MOSAIC, a post-training framework for improving the safety of agentic language models in multi-step tool-use settings. The key idea is to restructure agent reasoning into a Plan–Check–Act/Refuse loop, where explicit safety checking occurs before irreversible tool execution. The framework is trained using preference-based reinforcement learning with pairwise trajectory comparisons, enabling the model to learn temporally sensitive safety behaviors such as early refusal. Experimental results across several open-weight and frontier models show substantial reductions in harmful actions and privacy leakage while maintaining strong benign-task utility.

Strengths

1. The paper addresses a highly significant challenge in agentic AI: ensuring safe multi-step tool use in realistic environments where unsafe intermediate actions can cause irreversible harm.
2. The explicit Plan–Check–Act/Refuse decomposition is conceptually sound and practically meaningful, especially the treatment of refusal as a first-class learnable action rather than a post hoc safeguard. The integration of pairwise preference-based GRPO for trajectory-level safety alignment is also a strong technical contribution.
3. The evaluation spans multiple model families and benchmarks, demonstrating consistent safety improvements, strong transfer to privacy-sensitive domains, and favorable trade-offs between safety and utility.


Weaknesses

1. The reward signal relies heavily on GPT-4o-based trajectory judging, and the identified positional bias in preference comparisons raises concerns about hidden systematic biases in training supervision.
2. Although MOSAIC is designed for open-weight models, its current training pipeline depends on proprietary frontier systems for judging, which may constrain reproducibility and full self-hosted deployment.
3. Some experimental distinctions, especially between inference-only scaffolding and training-enhanced MOSAIC, are not sufficiently clear in the main results tables. Training details were initially sparse, and reproducibility concerns were raised before the repository clarification.


The rebuttal addressed most reviewer concerns satisfactorily. Questions regarding statistical significance, repository availability, and implementation details were largely resolved, leading multiple reviewers to explicitly acknowledge that their concerns had been fully addressed.

Overall, the paper presents a strong and well-executed contribution on an important problem, with clear novelty, solid empirical evidence, and meaningful practical impact.